# From Genetics to Neuroinflammation: The Impact of ApoE4 on Microglial Function in Alzheimer’s Disease

**DOI:** 10.3390/cells14040243

**Published:** 2025-02-07

**Authors:** Daniela Dias, Camila Cabral Portugal, João Relvas, Renato Socodato

**Affiliations:** i3S—Instituto de Investigação e Inovação em Saúde da Universidade do Porto, 4200-135 Porto, Portugal; daniela.dias@i3s.up.pt (D.D.); camila.portugal@ibmc.up.pt (C.C.P.); jrelvas@i3s.up.pt (J.R.)

**Keywords:** apolipoprotein E4 (ApoE4), microglia, Alzheimer’s disease (AD), neuroinflammation, amyloid-beta clearance, neurodegeneration, therapeutic strategies

## Abstract

Alzheimer’s disease (AD) is a debilitating neurodegenerative disorder marked by progressive cognitive decline and memory loss, impacting millions of people around the world. The apolipoprotein E4 (ApoE4) allele is the most prominent genetic risk factor for late-onset AD, dramatically increasing disease susceptibility and accelerating onset compared to its isoforms ApoE2 and ApoE3. ApoE4’s unique structure, which arises from single-amino-acid changes, profoundly alters its function. This review examines the critical interplay between ApoE4 and microglia—the brain’s resident immune cells—and how this relationship contributes to AD pathology. We explore the molecular mechanisms by which ApoE4 modulates microglial activity, promoting a pro-inflammatory state, impairing phagocytic function, and disrupting lipid metabolism. These changes diminish microglia’s ability to clear amyloid-beta peptides, exacerbating neuroinflammation and leading to neuronal damage and synaptic dysfunction. Additionally, ApoE4 adversely affects other glial cells, such as astrocytes and oligodendrocytes, further compromising neuronal support and myelination. Understanding the ApoE4–microglia axis provides valuable insights into AD progression and reveals potential therapeutic targets. We discuss current strategies to modulate ApoE4 function using small molecules, antisense oligonucleotides, and gene editing technologies. Immunotherapies targeting amyloid-beta and ApoE4, along with neuroprotective approaches to enhance neuronal survival, are also examined. Future directions highlight the importance of personalized medicine based on individual ApoE genotypes, early biomarker identification for risk assessment, and investigating ApoE4’s role in other neurodegenerative diseases. This review emphasizes the intricate connection between ApoE4 and microglial dysfunction, highlighting the necessity of targeting this pathway to develop effective interventions. Advancing our understanding in this area holds promise for mitigating AD progression and improving outcomes for those affected by this relentless disease.

## 1. Introduction

Alzheimer’s disease (AD) is a neurodegenerative condition that progresses over time, resulting in a decline in cognitive functions and memory loss. Recognized as the most common cause of dementia among individuals over 60 years old, AD poses a significant challenge to public health due to its increasing prevalence, a consequence of the global aging population [1]. First described by Alois Alzheimer in 1906, the extracellular deposition of amyloid-beta peptides (Aβ) and the accumulation of hyperphosphorylated tau proteins have been recognized as the hallmarks of AD [2,3]. The primary classification of AD is divided into two forms: early-onset Alzheimer’s disease (EOAD) and late-onset Alzheimer’s disease (LOAD). EOAD is often associated with mutations in genes such as amyloid precursor protein (APP), presenilin-1 (PSEN1), and presenilin-2 (PSEN2) [4,5]. On the flip side, LOAD accounts for about 95% of cases and has a multifactorial etiology that includes genetic risk factors, such as the ApoE gene [6,7].

Apolipoprotein E (ApoE), produced and secreted by astrocytes and microglia in the central nervous system (CNS), is essential for lipid metabolism, the maintenance of synaptic plasticity, and neuronal regeneration [8,9]. The polymorphic gene APOE encodes three common isoforms, ApoE2, ApoE3, and ApoE4, which differ by small amino acid substitutions at positions 112 and 158 [10,11]. The ε4 allele holds significant importance, representing 50–70% of AD cases and conferring a higher risk as well as an earlier onset compared to other isoforms [12,13]. In the brain, microglia comprise 10% of the total glial cell population, maintaining homeostasis by monitoring the neuronal environment and engaging in phagocytosis [14,15] However, if the ApoE4 allele is present, microglia adopt a pro-inflammatory profile, marked by increased expression of pro-inflammatory cytokines such as TNF-α, IL-1β, and IL-6 [16]. ApoE4 affects the function, morphology, and responsiveness of microglia to pathological stimuli. Therefore, developing effective therapeutics is vital and necessitates a comprehensive investigation into how these factors influence the pathological characteristics of AD, including amyloid plaque formation, tau pathology, and neuroinflammation.

This review examines the structural and functional characteristics of ApoE and its effects on brain health. We analyze the influence of ApoE4 on microglial function, focusing on recent findings regarding lipid metabolism dysregulation, impaired phagocytosis, and chronic neuroinflammation. Additionally, we discuss the interactions of ApoE4 with astrocytes and oligodendrocytes in neurodegeneration associated with AD. This review also presents a forward-looking perspective on emerging therapies targeting ApoE4 and microglia, including gene-editing technologies (e.g., CRISPR/Cas9), antisense oligonucleotides, and immune modulation. By synthesizing recent advancements and identifying research gaps, we aim to provide a comprehensive framework for future investigations and clinical strategies in AD.

## 2. ApoE: Structure, Function, and Role in the Brain

### 2.1. ApoE Structure and Isoforms

#### 2.1.1. Protein Structure: ApoE Gene Location, Sequence, and Molecular Structure

Apolipoprotein E (ApoE) is a 34 kDa glycoprotein composed of 299 amino acids encoded by the APOE gene on chromosome 19q13.2 [17]. The molecular structure of the protein comprises two main domains linked by a hinge region. The N-terminal domain, enclosing residues 1 to 191, forms a four-helix bundle and contains the receptor-binding region crucial for interactions with lipoprotein receptors like the low-density lipoprotein receptor (LDLR) and LDL receptor-related protein 1 (LRP1) [18]. The C-terminal domain, encompassing residues 216 to 299, is responsible for lipid binding and includes amphipathic α helices that interact with lipids, facilitating the formation of lipoprotein particles [10]. Structural studies using X-ray crystallography and nuclear magnetic resonance (NMR) revealed that the tertiary and quaternary structures of ApoE are influenced by its amino acid composition, particularly at positions 112 and 158. These positions are critical because they determine the isoform of ApoE and affect domain–domain interactions within the protein. In ApoE4, for example, the presence of arginine at position 112 facilitates the formation of a salt bridge between Arg61 in the N-terminal domain and Glu255 in the C-terminal domain. This interaction alters the conformation of the protein, making it more susceptible to cleavage and proteolytic aggregation [10,13]. Such structural variations have significant implications for the functional properties of the protein, particularly lipid metabolism and receptor binding.

#### 2.1.2. Isoforms: Differences Among ApoE2, ApoE3, and ApoE4; Impact on Protein Conformation and Function

The *APOE* gene is polymorphic, resulting in three common isoforms of ApoE, ApoE2, ApoE3, and ApoE4, encoded by the ε2, ε3, and ε4 alleles, respectively. These isoforms differ by single amino acid substitutions at positions 112 and 158. These substitutions lead to variations in the protein’s structure and function. In ApoE4, the arginine at position 112 interacts with Glu255 in the C-terminal domain, promoting a domain interaction that is not present in ApoE3 or ApoE2. This conformational change impacts the protein’s ability to interact with lipids and receptors. Compared to the other isoforms, ApoE4 has a higher affinity for very low-density lipoprotein (VLDL) receptors and less affinity for high-density lipoprotein (HDL) receptors. This difference influences how lipids are transported and cleared in the body.

ApoE2, with cysteine at both sites, has reduced binding affinity to LDL receptors due to the altered receptor-binding domain. This reduced affinity affects lipoprotein clearance and is associated with type III hyperlipoproteinemia. ApoE3, the most common isoform, is the neutral reference, displaying intermediate receptor affinity and normal lipid transport function.

The structural variations among the isoforms also impact their propensity to form aggregates and their stability. ApoE4 is more prone to misfolding and aggregation, which can contribute to pathological processes in the brain. The differences in structure and function among the isoforms underscore their distinct roles in lipid metabolism and their varying impacts on neurological health, particularly concerning Alzheimer’s disease risk and progression.

### 2.2. Function of ApoE in the Brain

#### 2.2.1. Lipid Transport: Role in Transporting Cholesterol and Other Lipids to Neurons

ApoE plays a pivotal role in the central nervous system by mediating the transport of cholesterol, phospholipids, and other essential lipids to neurons. Neurons have limited capacity for de novo cholesterol synthesis and rely on astrocyte-derived cholesterol for membrane maintenance, synapse formation, and repair [19]. ApoE-containing lipoprotein particles facilitate the delivery of these lipids by interacting with cell surface receptors on neurons.

The interaction with lipid transporters is critical for ApoE function. ABCA1, an ATP-binding cassette transporter A1, is responsible for the initial lipidation of ApoE, allowing for the generation of HDL-like particles in the CNS [20,21]. These particles may then interact with receptors such as LDLR and LRP1 on neuronal surfaces, mediating the endocytic uptake of lipids.

In individuals expressing the ApoE4 isoform, there is an impairment in ABCA1 transporter activity, leading to poorly lipidated ApoE particles. This results in reduced cholesterol efflux from astrocytes and accumulation of lipid droplets within cells, disrupting cellular homeostasis and contributing to neurodegeneration [8]. The decreased efficiency in lipid transport affects membrane fluidity and synaptic function, potentially exacerbating the accumulation of amyloid-beta peptides and promoting the formation of amyloid plaques.

#### 2.2.2. Neuronal Repair and Synaptic Plasticity: Mechanisms of Involvement in Neuronal Repair and Regeneration and Their Influence on Synaptic Formation and Maintenance

Beyond lipid transport, ApoE is instrumental in neuronal repair and synaptic plasticity. It facilitates the redistribution of lipids following neuronal injury, aiding in membrane repair and regeneration. By supplying essential lipids, ApoE supports the reformation of damaged neuronal structures and the maintenance of synaptic integrity [22].

ApoE influences synaptic plasticity by delivering cholesterol and phospholipids necessary for the growth and maintenance of dendritic spines and synapses. This lipid supply is crucial for synapse formation, remodeling, and function, which underpin learning and memory processes [18]. The interaction of ApoE with neuronal receptors can activate intracellular signaling pathways that are vital for neuronal survival and synaptic regulation, such as the MAPK and PI3K/Akt pathways.

The isoform-dependent effects of ApoE are evident in these processes. ApoE2 and ApoE3 promote neuroprotection by facilitating efficient lipid transport, supporting membrane integrity, and suppressing pro-inflammatory cytokine expression, thereby fostering a healthy neuronal microenvironment [12,23]. Conversely, ApoE4 impairs these functions. The altered lipidation of ApoE4 reduces its capacity to deliver lipids to neurons, hindering repair mechanisms and compromising synaptic plasticity. Furthermore, ApoE4 promotes a pro-inflammatory state, increasing the expression of cytokines, specifically TNF-α, IL-1β, and IL-6, which can lead to neuronal damage and exacerbate neurodegenerative processes [16,20].

ApoE4 also affects the clearance of amyloid-beta peptides. It impairs the ability of microglia and astrocytes to clear these peptides effectively, leading to their build-up and the formation of amyloid plaques, which disrupt synaptic function and contribute to cognitive decline [13,24,25]. The combined effects of impaired lipid transport, increased neuroinflammation, and reduced clearance of pathological proteins highlight the detrimental impact of ApoE4 on neuronal repair and synaptic plasticity.

## 3. Polymorphisms/Isoforms of ApoE and Their Effects on Alzheimer’s Disease

### 3.1. Genetic Variability of ApoE

#### 3.1.1. Polymorphisms: Description of APOE Gene Polymorphisms and Their Genetic Distribution

The APOE gene shows significant genetic variability due to polymorphisms that result in multiple isoforms with distinct functional properties [26,27]. The most extensively studied polymorphisms are the single nucleotide polymorphisms (SNPs) rs429358 and rs7412, which cause amino acid substitutions at positions 112 and 158. These variations yield the three common isoforms of ApoE: ApoE2, ApoE3, and ApoE4 [22,28,29]. The functional differences among these isoforms greatly influence lipid transport, amyloid-beta (Aβ) clearance, and neuroinflammation, thereby affecting an individual’s susceptibility to Alzheimer’s disease (AD).

The geographic distribution and prevalence of these alleles vary significantly across populations, shaped by evolutionary pressures, environmental factors, and historical migration patterns. ApoE3, the ancestral allele, is the most prevalent isoform worldwide, with a frequency of 65–70%, and is considered neutral concerning AD risk. ApoE2, which occurs in 5–10% of the population, is the least common isoform and is more frequent in certain groups. This isoform offers protection against AD but is linked with an increased risk of type III hyperlipoproteinemia. In contrast, ApoE4, which presents the highest genetic risk for late-onset AD, is most prevalent among population groups from Central Africa (45%), Oceania (37%), and Australia (26%) [30,31,32]. Meanwhile, populations of Asian and Native American ancestry show the lowest prevalence of ApoE4, correlating with reduced rates of AD [33].

Although the prevalence of ApoE4 accounts for some differences in AD risk across populations, this risk is also shaped by complex interactions with environmental factors, lifestyle, and other genetic modifiers. For example, non-Hispanic white individuals generally have a lower risk of developing AD compared to Black and African American populations; this disparity may reflect not only genetic differences but also social determinants of health. Factors such as socioeconomic status, access to healthcare, and higher rates of comorbidities in minority populations can significantly impact the progression and outcomes of AD. For instance, Black women have been reported to exhibit a higher vulnerability to AD compared to white women, further emphasizing the interaction between race and gender in AD risk.

Sex is another critical variable influencing the role of ApoE4 in AD progression. Women carrying the ApoE4 allele have a significantly higher risk of developing AD compared to men with the same genotype. This increased vulnerability is particularly noticeable after menopause, likely due to the decline in estrogen levels, which have neuroprotective effects. Estrogen regulates ApoE expression and lipid metabolism, and its decline may worsen the pathological impact of ApoE4 in women [29]. Moreover, female ApoE4 carriers display more severe amyloid-beta pathology, quicker cognitive decline, and a more significant accumulation of tau tangles than their male counterparts, emphasizing the interaction between sex and genetic risk in AD progression.

However, it is important to note that studies on ethnicity and its interaction with ApoE4 are often scarce or produce contradictory findings. This limitation emphasizes the need for expanded genetic and epidemiological research that involves diverse racial and ethnic groups. Current research disproportionately focuses on well-represented populations, often overlooking minority groups, which restricts our understanding of how ApoE4 interacts with specific genetic and environmental contexts. Addressing these gaps is crucial for identifying population-specific disease mechanisms and tailoring interventions to mitigate the inequalities underrepresented groups face.

Overall, the interplay between ethnicity, gender, and ApoE4 status adds significant complexity to AD risk and progression. Future studies should focus on including diverse cohorts to investigate how population-specific factors influence the effects of ApoE4 and to clarify existing disparities. Such efforts will be crucial for designing targeted interventions and advancing precision medicine strategies to combat AD effectively.

#### 3.1.2. Mechanistic Differences: How Each Isoform (ApoE2, ApoE3, ApoE4) Affects Brain Function

All ApoE isoforms affect brain function through distinct mechanisms, influencing lipid metabolism, neuroinflammation, and neuronal repair. ApoE2, a variant with cysteine residues at positions 112 and 158, has a reduced binding affinity for LDL receptors due to adjustments in the receptor-binding domain. Lipoprotein clearance is reduced, but so is amyloid-beta aggregation, which contributes to the variant’s neuroprotective effect [34]. ApoE2 promotes effective lipid transport and supports anti-inflammatory responses, contributing to neuronal protection and reducing the risk of AD.

The predominant isoform, ApoE3, serves as a neutral variant concerning the risk of AD. ApoE3 strictly maintains normal lipid transport and stable neuronal support functions, as well as the equilibrium between pro-inflammatory and anti-inflammatory dynamics to maintain the homeostatic function of neurons [11]. In contrast, apolipoprotein E4, with arginine residues at both positions, has a significant opposite effect. Its expression leads to an abnormal conformation that consequently results in the loss of lipids and cholesterol particles, contributing to the breakdown of membrane integrity [8]. Due to its impact on the interaction with cell surface receptors, an increase in the production of amyloid plaques is seen [10,35]. Furthermore, the expression of ApoE4 results in neuroinflammation as it induces the expression of pro-inflammatory cytokines, which can lead to cell damage by directly affecting microglial function [16]. This isoform impairs the blood–brain barrier, as it reduces the expression of an enzyme that protects the entry of toxic substances, which leads to synaptic dysfunction and behavioral deficits [36]. The mechanical differences between ApoE isoforms highlight the complex role of apolipoprotein E (ApoE) in CNS health and disease, highlighting the need to develop therapeutic methods targeting specific isoforms to treat AD.

### 3.2. Impact of ApoE4 on Alzheimer’s Disease

#### 3.2.1. Epidemiological Evidence: Studies Linking ApoE4 to Alzheimer’s Risk; Prevalence in Populations

Epidemiological studies have established a strong association between the ApoE4 allele and the increased risk of late-onset Alzheimer’s disease. Carriers of one copy of the ε4 allele have a 3 to 4 times higher risk of developing AD, while those that are homozygous for ε4 may have 12 to 15 times the likelihood compared to non-carriers [19,35]. The demographic variable of sex is fundamental because differences between men and women have been implicated in the risk of the pathology. The percentage of women with AD is higher than that of men, partly due to hormonal and gender differences and a longer life expectancy in females. Although there are indeed characteristics that lead to the consideration of the differences between men and women, these are not very consistent and may vary from study to study. Women carrying ApoE4 are more susceptible to AD than men carrying the same isoform due to hormonal factors such as the decline in estrogen levels during menopause [37,38]. Factors such as education, environmental habits, and cardiovascular health may modulate the risk associated with ApoE4. The genetic and environmental context is critical when assessing AD risk, because the small variations that exist at the ethnic, racial, or even lifestyle level of individuals can change the perspective regarding the role of isoforms, particularly ApoE4, in the predisposition to Alzheimer’s disease. One study showed that the risk of developing AD was differentiated depending on whether we were referring to white women or black women. Black women had a much higher risk percentage, suggesting that the vulnerability of these populations to the pathology was significant [39].

#### 3.2.2. Pathophysiological Mechanisms: Role in Amyloid-Beta Aggregation, Reduced Clearance, Tau Pathology, and Synaptic Dysfunction

ApoE4 is pathogenic in Alzheimer’s disease via several interconnected mechanisms. One of these is its role in the aggregation and clearance of amyloid-beta. ApoE isoforms vary in their effectiveness in eliminating Aβ peptides from brain interstitial fluid. ApoE4 is less effective at clearing Aβ due to a low removal rate of this peptide, which causes it to aggregate and lead to the formation of amyloid plaques [40,41]. Additionally, ApoE4 may more easily promote Aβ aggregation in neurotoxic oligomers and fibrils, further exacerbating amyloid pathology. 

Regarding tau pathology, ApoE4 promotes hyperphosphorylation of tau proteins, which triggers the generation of neurofibrillary tangles (NFTs) [6]. ApoE4 incentivizes the accumulation of these NFTs through its ability to deregulate kinases and phosphatases involved in the tau phosphorylation/dephosphorylation cycle, including glycogen synthase kinase 3 beta (GSK-3β) and protein phosphatase 2A (PP2A) [42,43]. ApoE4 microglia induce a pro-inflammatory microenvironment that may also contribute to tau pathology by releasing cytokines that affect its metabolism.

ApoE4 compromises synaptic function and plasticity. It impairs synaptic plasticity by disrupting lipid transport and membrane repair mechanisms, which are linked to maintaining synaptic integrity [13]. The integrity of the blood–brain barrier (BBB) is disrupted by APOE4, allowing neurotoxic substances such as cyclophilin A to enter the CNS, potentiating neuronal dysfunction [36]. These pathophysiological mechanisms demonstrate in detail the effects of ApoE4 on the progression of neurodegeneration observed in AD, highlighting the central role of this isoform in the predisposition to the disease. It is important to include information regarding 5XFAD mouse models in AD research. This model presents a relatively early onset of amyloid pathology compared to other models used. The presence of ApoE4 contributes to exacerbated clinical outcomes in the 5XFAD models, as together they exacerbate some of the pathological clinical aspects of AD, such as an increase in amyloid-beta load, a significant progression of cognitive decline, and impaired neuronal communication [44].

#### 3.2.3. Comparative Analysis: Differences in Disease Progression and Severity Among ApoE2, ApoE3, and ApoE4 Carriers

Comparative studies among carriers of different ApoE isoforms have shown distinct patterns in the progression and severity of AD, highlighting the varying effects of ApoE2, ApoE3, and ApoE4 on disease pathology (Table 1). These differences stem from the unique structural and functional properties of each isoform, which influence crucial processes such as lipid metabolism, amyloid-beta (Aβ) clearance, and neuroinflammation.

The ApoE2 isoform is linked to protective effects against AD. It fosters an anti-inflammatory environment that enhances microglial phagocytosis of neurotoxic proteins and promotes the efficient clearance of Aβ, thus reducing the accumulation of amyloid plaques. ApoE2 also aids lipid transport and supports synaptic health, contributing to a neuronal environment that is less vulnerable to damage [45]. Its lower binding affinity to Aβ, compared to ApoE3 and ApoE4, further diminishes the risk of plaque formation [12]. Consequently, carriers of ApoE2 typically experience a delayed onset of AD symptoms and demonstrate milder disease progression, with less amyloid deposition and better-preserved cognitive function, even if the disease manifests [34]. The cumulative impact of these protective mechanisms positions ApoE2 as the most favorable isoform in relation to AD risk and progression.

ApoE3, being the most common isoform, is regarded as the “neutral” reference allele in AD research. While it does not provide significant protection like ApoE2, it also does not exhibit the harmful effects associated with ApoE4. ApoE3 maintains balanced lipid transport and a steady inflammatory profile, which supports neuronal health and homeostasis. Notably, ApoE3 can manage microglial responses to damage without compromising neuronal function or triggering excessive inflammation. Its moderate binding affinity to Aβ allows for effective plaque clearance without overactivating the inflammatory cascade. This equilibrium explains why ApoE3 carriers show intermediate risk and disease severity compared to those carrying ApoE2 and ApoE4.

Conversely, the ApoE4 isoform markedly accelerates AD progression and heightens disease severity. ApoE4 disrupts microglial homeostasis, leading to diminished phagocytic capacity and uncontrolled inflammatory responses. Microglia expressing ApoE4 adopt a pro-inflammatory phenotype, marked by the release of cytokines such as TNF-α, IL-1β, and IL-6, which intensify neuroinflammation and promote neuronal damage [45]. In addition, ApoE4 modifies lipid metabolism, resulting in inadequate lipid transport and an accumulation of lipid droplets within microglia, further impairing their ability to clear Aβ [34].

The distinct structural properties of ApoE4 also affect its binding affinity to Aβ. In contrast to ApoE2 and ApoE3, ApoE4 binds more readily to Aβ, leading to the aggregation of toxic oligomers and fibrils that form senile plaques [46]. This heightened tendency for aggregation, combined with impaired microglial clearance, results in a greater amyloid burden among ApoE4 carriers. Additionally, ApoE4 worsens tau pathology by promoting hyperphosphorylation of tau proteins, resulting in neurofibrillary tangles that further contribute to neuronal dysfunction and cognitive decline [34,47].

ApoE4 also exhibits an age-dependent effect on disease progression, with older carriers experiencing more rapid cognitive decline and greater neurodegeneration. This may result from cumulative damage caused by chronic inflammation, oxidative stress, and the ongoing buildup of amyloid plaques and tau tangles over time. Furthermore, ApoE4 impairs synaptic plasticity and accelerates the loss of dendritic spines, further impairing neuronal communication and worsening cognitive deficits.

The stark differences between ApoE2, ApoE3, and ApoE4 underscore the unique and complex role of ApoE4 in driving AD pathology. Understanding these distinctions is crucial for developing personalized therapeutic strategies that address the specific pathways disrupted by each isoform. For instance, therapies aimed at reducing the pro-inflammatory effects of ApoE4, enhancing its lipid transport functions, or mitigating its role in Aβ aggregation could offer isoform-specific benefits for those who carry ApoE4. These insights emphasize the importance of customizing interventions to tackle the distinct contributions of each ApoE variant to the development and progression of AD.

**Table 1 cells-14-00243-t001:** Differences observed between apolipoprotein E (ApoE) isoforms. The characteristics of the different isoforms are key to understanding the direct influence of each one on brain health and Alzheimer’s disease (AD).

**Isoform**	**Composition**	**Lipid Function**	**Interaction with Aβ**	**Metabolic Effects**	**Implications in Neuroinflammation**	**References**
ApoE2	Cys112, Cys158	Affinity high for LDL and low affinity for HDL	Reduced Aβ binding capability	High triglyceride and cholesterol levels	May exacerbate tau pathology	[34,48]
ApoE3	Cys112, Arg158	Preference interaction with HDL	Moderate Aβ-binding capacity	Lipid balance in the CNS	Considered neutral with respect to neuronal connectivity	[11,48]
ApoE4	Arg112, Arg158	Increased affinity for VLDL	Increased Aβ binding capacity	Associated with atherosclerosis and dyslipidemia	Increased Aβ and tau load, impairing cognitive function	[48,49]

## 4. Interaction Between ApoE4 and Microglia

### 4.1. Functional Role of Microglia in the Healthy and Diseased Brain

#### 4.1.1. Homeostatic Functions: Immune Surveillance, Synaptic Pruning, Response to Injury

Microglia are CNS-resident immune cells originating from yolk sac progenitors, which represent approximately 10% of the total glial population [14,15]. In a healthy brain context, microglia play crucial roles in maintaining homeostasis through brain surveillance, synaptic pruning, and injury response. They continually scan the CNS environment, extending and retracting its processes to detect injuries [50]. This surveillance process allows them to quickly and effectively respond to pathogens and damaged neurons, initiating a phagocytic response and releasing cytokines to adjust to the immune response [51]. Another vital function of microglia is synaptic pruning, especially during development, but also in the adult brain. By engulfing and eliminating very strong or extremely weak synaptic connections, microglia refine neural circuits to enhance communication and function between neurons [52]. This process of synaptic pruning is critical for enhancing learning, memory retention, and cognitive stability. In reaction to trauma, microglia change their phenotype from vigilant to active, moving to the site of damage to eliminate debris and release components that support tissue repair and regeneration. CNS integrity is regulated by the ability of microglia to perform these homeostatic functions. However, dysregulation of this microglial activity may result in inappropriate immune responses and be a driver for the pathogenesis of AD.

#### 4.1.2. Microglia Activation States

Microglial activation is now widely recognized as existing on a continuum of functional states, with their phenotypic profiles influenced by the complex and dynamic microenvironment of the CNS. These states vary from pro-inflammatory and neurotoxic to anti-inflammatory and neuroprotective, and they also include intermediate or mixed activation states that combine elements of both ends of the spectrum. This emerging perspective emphasizes the nuanced roles of microglia in health and disease, highlighting their ability to adapt to different physiological and pathological conditions [34]. Unlike a rigid dichotomy, this spectrum represents a fluid and context-dependent response to stimuli such as infections, injury, aging, and genetic factors like ApoE4.

However, for simplicity and to foster a more accessible and holistic understanding of ApoE4’s impact within this review, we adopt the classical dichotomy of pro-inflammatory (“M1”) and anti-inflammatory (“M2”) profiles. While this binary framework is an oversimplification, it provides a clear and structured approach for discussing the effects of ApoE4 on microglial behavior and its downstream consequences in AD pathology.

The pro-inflammatory (often referred to as “M1”) phenotype is typically activated in response to pathogens or cellular damage, initiating an inflammatory response to protect the CNS. Microglia in this state release pro-inflammatory cytokines, such as tumor necrosis factor-alpha (TNF-α), interleukin-1 beta (IL-1β), interleukin-6 (IL-6), and reactive oxygen species (ROS), which are essential for defense [53,54]. However, prolonged or excessive activation of this phenotype can lead to chronic neuroinflammation, worsening neuronal damage and contributing to the progression of neurodegenerative diseases like AD.

In contrast, the anti-inflammatory (“M2”) phenotype is associated with the release of cytokines such as interleukin-4 (IL-4), interleukin-10 (IL-10), and transforming growth factor beta (TGF-β), which support neuronal survival, suppress inflammatory processes, and promote tissue repair [55,56]. This state is essential for maintaining homeostasis and resolving inflammation following injury or stress.

It is vital to acknowledge that microglial polarization is highly dynamic and influenced by a variety of factors, including stress, aging, and genetic predispositions such as ApoE4. These factors drive microglia to exist in mixed or intermediate states, reflecting the complexity of their functional roles in the CNS. By framing microglial activation within this dichotomy, we aim to provide a simplified yet illustrative context for understanding the influence of ApoE4 on microglial behavior and its broader implications for AD pathology. Ultimately, achieving a balance between these states is key to maintaining brain tissue homeostasis and preventing disease progression.

### 4.2. Impact of ApoE4 on Microglial Function

#### 4.2.1. Phenotypic Changes: Altered Activation States and Behavior in the Presence of ApoE4

In the presence of the ApoE4 isoform, microglia show significant changes in their functional states (Figure 1), characterized by increased production of pro-inflammatory cytokines, decreased phagocytic ability, and impaired chemotactic and migratory behavior [57,58,59]. These phenotypic changes are closely tied to ApoE4-induced alterations in gene expression profiles. Specifically, ApoE4 enhances the expression of genes linked to inflammation while reducing those associated with microglial homeostasis and repair (Table 2). This dual effect causes microglia to adopt a pro-inflammatory state, which compromises their ability to clear pathological proteins like amyloid-beta (Aβ) and other cellular debris [19,26]. One of the key features of ApoE4-expressing microglia is the disruption of lipid metabolism, leading to the build-up of intracellular lipid droplets (Table 2). The accumulation of lipid droplets subsequently triggers inflammatory signaling pathways that intensify oxidative stress and impair cellular function [7,60]. This metabolic dysfunction not only limits the ability of microglia to maintain a balanced environment but also exacerbates the neuroinflammatory processes associated with AD). Furthermore, ApoE4 affects microglial morphology, often resulting in shorter and less branched processes, which further reduces their capacity for environmental monitoring and debris clearance [57,58]. These cumulative effects of ApoE4 push microglia into a state of chronic activation, perpetuating an environment of inflammation, oxidative stress, impaired phagocytic activity, and neuronal damage, which considerably contributes to the pathogenesis and progression of AD (Figure 1). Targeting these phenotypic changes through strategies aimed at modifying ApoE4-induced gene expression and lipid metabolism may help restore microglial function and mitigate their detrimental impact on AD pathology.

#### 4.2.2. Cytokine Production: Changes in Inflammatory Cytokine Profiles

A hallmark feature of ApoE4-expressing microglia is their increased production and release of pro-inflammatory cytokines, including tumor necrosis factor-alpha (TNF-α), interleukin-1 beta (IL-1β), interleukin-6 (IL-6), and interferon-gamma (IFN-γ) [16]. This dysregulated cytokine release establishes a cycle of sustained neuroinflammation, where inflammatory mediators activate nearby microglia and astrocytes, amplifying the inflammatory response (Table 2). Over time, this chronic inflammation creates an unfavorable environment for neuronal survival and synaptic integrity.

The molecular mechanisms underlying this increased cytokine production include the activation of the NF-κB signaling pathway, a key regulator of inflammatory responses. ApoE4 enhances NF-κB activity, leading to the transcriptional upregulation of genes that encode pro-inflammatory cytokines and chemokines [28,61]. This abnormal activation not only drives microglial dysfunction but also suppresses the anti-inflammatory response, which is essential for resolving inflammation and promoting repair. Furthermore, this prolonged pro-inflammatory state fosters the recruitment of peripheral immune cells into the CNS, facilitated by blood–brain barrier (BBB) disruption, thereby worsening the inflammatory response.

In addition to cytokine production, ApoE4 microglia contribute to the generation of reactive oxygen species (ROS) and reactive nitrogen species (RNS), which increase oxidative stress and neuronal damage [4,56]. Oxidative stress further activates microglia and perpetuates the release of cytokines, creating a feedback loop that sustains inflammation and exacerbates AD pathology.

Therapeutic strategies that target these inflammatory pathways, such as inhibitors of NF-κB or agents that promote an anti-inflammatory phenotype, show promise for breaking this cycle and restoring a more balanced microglial response.

#### 4.2.3. Phagocytic Activity: Impaired Clearance of Amyloid-Beta and Other Debris

Microglial phagocytosis is a critical process for maintaining central nervous system (CNS) homeostasis by removing pathological proteins such as amyloid-beta (Aβ) and cellular debris. This function ensures the clearance of neurotoxic elements and prevents the accumulation of damaging aggregates. However, ApoE4 severely impairs this essential function, resulting in a significant reduction in microglial phagocytic efficiency and their ability to migrate toward Aβ plaques (Table 2). This impairment leads to the accumulation of Aβ plaques, which exacerbates AD pathology and contributes to disease progression [12,57].

Compared to microglia expressing ApoE2 or ApoE3, ApoE4-expressing microglia demonstrate a significant decrease in phagocytic activity, which is mechanistically linked to molecular-level changes. ApoE4 downregulates the expression of critical genes involved in phagocytosis, including the purinergic receptor P2RY12 and the Triggering Receptor Expressed on Myeloid Cells 2 (TREM2).

P2RY12 plays a vital role in microglial chemotaxis, allowing microglia to detect and migrate toward areas of injury or amyloid deposition. However, in ApoE4-expressing microglia, the reduced expression of P2RY12 severely compromises this chemotactic response, leaving amyloid plaques and other pathological features unaddressed [51,59].

Similarly, TREM2 is essential for microglial activation and the initiation of phagocytic responses to lipid debris and amyloid deposits. The dysregulation of TREM2 in ApoE4 microglia further undermines their ability to clear Aβ plaques, as TREM2-mediated signaling pathways are critical for cytoskeletal rearrangement and lysosomal function, both crucial for effective phagocytosis [62,63]. The loss of TREM2 activity also diminishes the microglial capacity to respond to inflammatory signals, thereby hindering the resolution of neuroinflammation and contributing to a toxic environment in the brain. The consequences of impaired phagocytosis extend beyond amyloid clearance. The accumulation of unresolved Aβ and other debris serves as a chronic stimulus for microglial activation, creating a self-perpetuating cycle of inflammation. Persistent activation results in increased production of pro-inflammatory cytokines and reactive oxygen species (ROS), both of which worsen neuronal damage and synaptic dysfunction. Over time, this pro-inflammatory state leads to neuronal death, synaptic loss, and cognitive decline, which are hallmark features of AD pathology.

In addition to the direct effects on Aβ clearance, the phagocytic deficits observed in ApoE4 microglia also disrupt the clearance of other cellular debris, including apoptotic cells and damaged organelles. This inability to maintain cellular homeostasis causes the release of additional danger-associated molecular patterns (DAMPs), which further activate surrounding microglia and astrocytes. This amplifies the neuroinflammatory response and deteriorates BBB integrity, enabling peripheral immune cells to infiltrate the CNS and aggravate the inflammatory cascade [51,59].

Interestingly, recent studies suggest that ApoE4-induced phagocytic deficits may also impact microglial interactions with synapses. Microglia play a critical role in synaptic pruning, a process that eliminates weak or unnecessary synapses to refine neural circuits. Dysregulated phagocytosis in ApoE4 microglia may lead to improper pruning, resulting in the loss of functional synapses and contributing to impaired synaptic plasticity and cognitive decline observed in AD [57,62].

Addressing the phagocytic deficits in ApoE4-expressing microglia is crucial for breaking this cycle of dysfunction. Strategies such as activating TREM2 signaling or targeting pathways involved in cytoskeletal dynamics and lysosomal function may restore microglial capacity to clear Aβ and other pathological debris. Additionally, therapeutic approaches aimed at enhancing P2RY12-mediated chemotaxis could improve the ability of microglia to migrate toward and respond to pathological stimuli. Together, these interventions hold the potential to reduce the amyloid burden, resolve neuroinflammation, and mitigate the neurodegenerative effects of ApoE4 in AD.

**Table 2 cells-14-00243-t002:** Direct causal pathways linking ApoE4 to microglial dysfunction.

**Causal Route**	**Biological Mechanism**	**Impact on Microglia**	**References**
Chronic inflammation	Continuous activation of ApoE4 microglia results in excessive release of pro-inflammatory cytokines	Microglia do not act effectively to remove proteins, culminating in the accumulation of amyloid-beta	[16]
Oxidative stress	ApoE4 increases reactive oxygen species (ROS)	Compromised microglial function due to cellular damage associated with ROS, leads to neuronal death	[4,56]
Lipid homeostasis dysfunction	Lipid homeostasis is altered in the presence of the E4 allele through reduced interaction with receptors (LRP1)	Improper lipid transport that results in harmful immune responses that compromise neuronal health	[8]
Change in receiver signaling	Critical receptors such as TREM2 are affected by APOE4	Decreased ability of microglia to act in response to cellular injury or damage	[64]
Altered phagocytic capacity	ApoE4 impairs the signaling of receptors that potentiate Aβ phagocytosis	Removal of protein aggregates does not occur, leading to build-up that results in the formation of amyloid plaques	[62,63]

This table demonstrates some of the pathways that connect the ApoE4 isoform with dysregulated microglial function. For each of the causal pathways, the biological mechanisms and impacts at the microglial level are described, demonstrating that ApoE4 contributes to processes related to chronic inflammation and changes in phagocytosis, among others. These mechanisms lead to potentially adverse consequences for neuron health and the progression of neurodegenerative diseases such as Alzheimer’s disease (AD).

### 4.3. ApoE4 and Other Glial Cells

#### 4.3.1. Astrocytes: Effects on Astrocytic Support Functions and Neuroinflammatory Responses

Astrocytes play a critical role in maintaining CNS homeostasis by providing metabolic support to neurons, regulating neurotransmitter levels, and preserving the integrity of the BBB [65]. They are also the primary producers of ApoE in the brain. However, the presence of ApoE4 alters astrocytic function, transforming these supportive cells into contributors to pathology. One major effect of ApoE4 is its disruption of lipid metabolism in astrocytes [9]. This isoform impairs cholesterol synthesis and transport by decreasing the efficiency of interaction with ATP-binding cassette transporters (e.g., ABCA1), leading to the accumulation of intracellular cholesterol. This lipid dysregulation compromises membrane integrity, reduces synaptic function, and deprives neurons of essential lipid resources necessary for repair and plasticity [7,63,66]. Astrocytic reactivity, characterized by increased expression of glial fibrillary acidic protein (GFAP), is another hallmark of ApoE4-associated dysfunction. Elevated plasma concentrations of GFAP correlate with tau hyperphosphorylation and neurofibrillary tangle formation, establishing astrocytic reactivity as an early marker of AD [67]. This reactivity not only exacerbates neuroinflammation but also facilitates pathological interactions between Aβ and tau, amplifying the severity of tau pathology [68]. Additionally, ApoE4-induced astrocytic dysfunction worsens the inflammatory response through cross-talk with microglia. Reactive astrocytes release pro-inflammatory cytokines, such as IL-1β and TNF-α, that activate neighboring microglia, creating a feedback loop that perpetuates chronic inflammation. These synergistic interactions further compromise neuronal health and accelerate neurodegeneration. Studies suggest that selectively removing ApoE4-expressing astrocytes can reduce microglial reactivity and provide cerebrovascular protection, highlighting the therapeutic potential of targeting astrocytic dysfunction [60]. Similarly, the deletion of ApoE4-expressing microglia has been shown to activate astrocytes, emphasizing the complex interplay between these glial cells in the context of ApoE4 [45]. Understanding these interactions is crucial for developing interventions aimed at restoring astrocytic function and CNS homeostasis.

#### 4.3.2. Oligodendrocytes: Impact on Myelination and White Matter Integrity

Oligodendrocytes are essential for myelinating axons in the CNS, facilitating the efficient conduction of action potentials. However, ApoE4 adversely affects oligodendrocyte function, diminishing their capacity to produce and maintain myelin. This results in disruptions in white matter integrity and impaired neuronal communication, which are linked to the cognitive deficits characteristic of AD [34,69].

One major consequence of ApoE4 for oligodendrocytes is the dysregulation of lipid metabolism, which is crucial for synthesizing myelin membranes. ApoE4 alters cholesterol biosynthesis and promotes lipid accumulation in oligodendrocytes, hampering their ability to produce and sustain the myelin sheath [35]. This dysfunction is further worsened by ApoE4-induced inflammatory signals from microglia and astrocytes that disrupt the homeostatic environment essential for oligodendrocyte function.

Oligodendrocyte precursor cells (OPCs), responsible for the regeneration and remyelination of axons, are also negatively impacted by ApoE4. Pathological changes associated with ApoE4 impair OPC proliferation, differentiation, and migration, obstructing the repair of damaged myelin [70]. Furthermore, recent evidence suggests that oligodendrocytes themselves may contribute to Aβ production in AD, with an increased number of oligodendrocytes expressing APP and processing it into Aβ. This finding implicates oligodendrocytes as active participants in amyloid pathology, adding another layer to their role in disease dynamics [71].

Therapeutic strategies aimed at restoring oligodendrocyte function and white matter integrity hold significant promise. These approaches include stem cell therapies to regenerate OPCs, lipid-targeting treatments to correct metabolic dysfunction, and anti-inflammatory interventions to protect oligodendrocytes from secondary damage. By addressing the specific vulnerabilities of oligodendrocytes and OPCs in the context of ApoE4, these therapies could mitigate cognitive decline and improve outcomes in AD.

## 5. Microglia, ApoE4, and Alzheimer’s Disease Pathology

### 5.1. Neuroinflammation and Microglial Activation

#### 5.1.1. Mechanisms of Neuroinflammation: ApoE4-Induced Pathways Leading to Chronic Inflammation

Neuroinflammation, one of the most prominent pathological hallmarks of Alzheimer’s disease (AD), is heavily modulated by the ApoE4 allele. ApoE4 modifies microglial function through various means, leading to chronic neuroinflammatory states. Altered gene expression associated with immunological responses leads to the upregulation of pro-inflammatory genes such as IL-1β, TNF-α, and NLRP3 inflammasome components [37,40] The E4 allele enhances the activation of the NF-κβ signaling pathway and mitogen-activated protein kinase (MAPK), pathways that govern inflammatory responses [61,72]. Increased NF-κβ activation transcriptionally promotes pro-inflammatory cytokines and chemokines, which exacerbate and perpetuate inflammation. The protein components of the NLRP3 inflammasome, such as the NLRP3 and ASC proteins, are positively impacted by the NF-κβ signaling pathway. As a result, once activated, the NLRP3 inflammasome which activates caspase-1, and possibly leads to cell death, is expressed at higher levels [73]. The activation of the ApoE4 microglia induces the accumulation of lipid droplets, which in turn triggers the activation of the inflammasome and the release of inflammatory mediators [7,74]. Indeed, the ApoE4 isoform itself disrupts its binding to TREM2, a receptor involved in the activation of microglia in response to lipid and amyloid stimuli [8,63]. This defect leads to an inflammatory brain microenvironment and impairs microglial phagocytosis. In addition, the increased expression of cyclophilin A in pericytes due to E4 promotes blood–brain barrier disruption, facilitating the infiltration of peripheral immune cells into the CNS and exacerbating neuroinflammation [36]. All these activities generate an inflammatory environment in the CNS, which potentiates the death and dysfunction of neurons that underlie neuropathological conditions.

#### 5.1.2. Microglial Response to Amyloid Plaques: Role in Plaque Formation and Progression

Microglia play a dual position in the dynamics of amyloid plaques, being both protective and pathological. In the initial stages of Alzheimer’s disease, microglia attempt to eliminate Aβ peptides by phagocytosis, which delays plaque formation [14]. In some cases, they can create physical barriers around the plates, isolating them and preventing the spread of toxic species [8]. In individuals with ApoE4, microglia see their ability to perform phagocytosis decreased, in addition to impaired chemotaxis about Aβ plaques, which helps in the growth and in a greater number of amyloid-beta plaques. The microglia recognize these plaques and become active to clear away the toxic elements [19,26]. However, this attempt may lead to the chronic activation of microglia, creating a cycle of release of inflammatory mediators that promote greater Aβ aggregation and neurotoxicity. Specific microglia subpopulations, including disease-associated microglia (DAM) and inflammation-related microglia (IRM), become more common with E4 expression [37], meaning they express fewer homeostatic genes and exhibit a strengthened neuroinflammatory profile. The interconnection between ApoE4 and TREM2 modulates the microglial response [64]. While the deletion of the ApoE4 isoform reduces Aβ pathology and promotes protection through the microglial barriers around the plaques, the loss of TREM2 does not confer this protection by not grouping the microglia and aggravates amyloid pathology [8]. Understanding the microglial response to amyloid pathology is necessary for developing strategies designed to increase Aβ clearance and decrease neuroinflammation in Alzheimer’s disease.

### 5.2. Neuronal Loss and Synaptic Dysfunction

#### 5.2.1. Neurodegeneration: Mechanisms by Which ApoE4-Expressing Microglia Contribute to Neuronal Loss

Microglia expressing ApoE4 contribute to neurodegeneration through many interrelated mechanisms. Chronic neuroinflammation activity from these microglia leads to the continuous secretion of pro-inflammatory cytokines and chemokines and results in a toxic environment for neuronal cells, promoting processes involved in cell apoptosis and necrosis [16,51]. The aberrant production of reactive oxygen species (ROS) and reactive nitrogen species (RNS) results in oxidative stress which impairs neuronal DNA, proteins, and lipids [4,56]. Microglia can also contribute to synaptic dysfunction via dysregulated synaptic pruning. ApoE4 microglia can also contribute to synaptic dysfunction by eliminating synaptic elements inappropriately [27]. This process occurs because the activation of the complement cascade can signal microglia to eliminate unnecessary synapses; however, this excessive elimination can lead to the loss of essential synapses. Brain-derived neurotrophic factor (BDNF) is a neurotrophin that is essential for both the survival of neurons and the formation of new synapses. In individuals carrying the ApoE4 isoform, the low BDNF availability in the brain exacerbates the negative influence of microglia and consequently creates an unfavorable neuronal environment that diminishes support for neuronal viability and plasticity. Collectively, these mechanisms illustrate the contribution of ApoE4 microglia in neuronal loss and the pathogenesis of AD.

#### 5.2.2. Synaptic Health: Impact on Synaptic Density and Function; Cognitive Implications

Synaptic integrity is crucial for cognitive function, and ApoE4 affects synaptic health through several methods. Some studies have found ApoE4 carriers to have a decreased dendritic column density and decreased synaptic marker expression [75]. The dendritic column is fundamental in the communication between various nerve cells, and a reduction in its density creates an imbalance in the ability of neurons to form and maintain synapses. ApoE4 impairs synaptic plasticity by inhibiting long-term potentiation (LTP), a cellular mechanism underlying learning and memory [76]. Dysregulated lipid transport associated with ApoE4 leads to extracellular accumulation of lipoproteins, which impairs neuronal activity [52,63]. In addition, inadequate or excessive synaptic pruning by activated microglia results in weak neuronal networks and loss of synapses [27]. These changes are responsible for cognitive decline in patients with Alzheimer’s disease, emphasizing the need for and importance of synaptic health in therapeutic strategies.

### 5.3. Interplay with Other Alzheimer’s Pathological Features

#### 5.3.1. Tau Pathology: Interaction with Tau Proteins and Neurofibrillary Tangles

Multiple dysregulated mechanisms mediate the interaction of ApoE4, microglia, and tau pathology. ApoE4 increases tau hyperphosphorylation and the subsequent development of neurofibrillary tangles (NFTs) that make neurons dysfunctional and promote neuronal death [28,76]. Under normal physiological conditions, glycogen synthase kinase 3 beta (GSK-3β) and cyclin-dependent kinase 5 (CDK5) regulate tau phosphorylation. However, the influence of ApoE4 on these two enzymes creates hyperphosphorylation of the tau protein. Exacerbating tau pathology, pro-inflammatory microglia secrete cytokines that modulate tau kinases and phosphatases [53]. In addition, microglia can internalize and release tau aggregates, spreading them throughout the brain and creating positive feedback that accentuates neurodegeneration [77]. In some study models, decreased ApoE4 levels reduced tauopathy and neuronal degeneration [78]. Thus, understanding the role of ApoE4 in tau pathology is important to develop interventions aimed at combating this aspect of Alzheimer’s disease (AD).

#### 5.3.2. Other Neurodegenerative Processes: Contribution to Overall Brain Atrophy and Connectivity Loss

In addition to amyloid and tau pathologies, ApoE4 together with microglia is associated with several neurodegeneration processes. Chronic microglial activation leads to an excessive release of pro-inflammatory cytokines and neurotoxic factors, promoting large-scale neuronal death and contributing to brain atrophy. Studies involving magnetic resonance imaging have shown that in the hippocampal and cortical regions of ApoE4 patients, there is a decrease in brain volume, which correlates with cognitive impairments [1].

The negative impact of ApoE4 on oligodendrocyte function and myelination results in demyelination and leads to the disruption of neural connectivity [35]. This loss affects the aspects involved in cognitive function, including memory, attention, and executive function.

The collaborative effect of synaptic loss, neuronal death, and white matter dissociation results in the profound functional decline noted in individuals with Alzheimer’s disease. As such, it is important to prioritize therapeutic approaches that aim to address various facets of neurodegeneration.

## 6. Therapeutic Potential and Future Perspectives

### 6.1. Current Therapeutic Approaches Targeting ApoE4 and Microglia

#### 6.1.1. Modulating ApoE4 Function: Small Molecules, Antisense Oligonucleotides, and Gene Editing

There are several promising therapeutic pathways available to target the function of ApoE4 to combat its detrimental effects on Alzheimer’s disease (Table 3). Small molecules can be engineered to alter the ApoE4 structure and function so that they can be converted to the ApoE3 or ApoE2 isoforms, which are less harmful, inhibiting their adverse impact on microglia and improving synaptic function [79,80]. These molecules act by stabilizing the protein structure, preventing the domain interactions that induce pathogenic conformational shifts. Antisense oligonucleotides (ASOs), considered synthetic polymers, are part of the first line of therapeutic strategies aimed at neurodegeneration processes. Synthetic nucleic acids bind to the messenger RNA (mRNA) of the apoE4 isoform, promoting its degradation and expression in the brain [66,81,82]. By lowering ApoE4 levels, these antisense oligonucleotides can reduce amyloid-beta accumulation and neuroinflammation and improve cognitive decline. Finally, gene-editing technologies, such as CRISPR/Cas9, have the potential to correct ApoE4 mutations at the genomic level. This approach facilitates the editing of the ApoE gene to convert the ε4 allele to ε3 or ε2, thus reversing the neurodegenerative effect associated with ApoE4 [83]. Although they are promising therapeutics, emerging challenges in specific delivery methods and ethical considerations would need to be addressed before their clinical application [66].

#### 6.1.2. Targeting Microglial Activation: Anti-Inflammatory Drugs and Immune Modulators

Modulating microglial activation to reduce neuroinflammation is another therapeutic strategy. Anti-inflammatory drugs, including nonsteroidal anti-inflammatory drugs (NSAIDs), can reduce brain inflammation by decreasing microglial activation and lowering pro-inflammatory cytokine release, thereby protecting neuronal and cognitive functions [79,84]. However, clinical trials have yielded mixed results, and further research is needed to optimize their efficacy and safety.

Immune modulators targeting receptors such as TREM2 can promote the anti-inflammatory M2 phenotype in microglia. Monoclonal antibodies that activate TREM2 enhance microglial phagocytosis, reduce amyloid plaques, and decrease neurofibrillary tangles [62,85,86,87]. These approaches aim to restore microglial homeostasis and improve their capacity to clear pathological proteins.

Combination therapies that include immune modulators, anti-inflammatory agents, and lifestyle adjustments may offer synergistic benefits in managing Alzheimer’s disease by targeting multiple aspects of microglial dysfunction.

### 6.2. Emerging Therapies and Research Directions

#### 6.2.1. Gene Editing Technologies: CRISPR/Cas9 Approaches to Correct ApoE4 Mutations

Gene-editing technologies like CRISPR/Cas9 hold significant promise for correcting ApoE4 mutations and potentially contributing to a cure for Alzheimer’s disease [88]. By precisely targeting and modifying the APOE gene, the researchers aim to convert the ε4 allele into the ε3 or ε2 allele, thus eliminating the source of ApoE4’s detrimental effects [83]. Studies conducted in vitro have confirmed the feasibility of this approach, showing reduced pathological features in cell models after some gene editing [89,90,91].

Challenges remain in developing safe and efficient delivery mechanisms to specifically target neuronal and glial cells in the brain, in off-target minimization, and in addressing ethical considerations regarding genetic modification. Continued advances in gene editing techniques and delivery systems are essential to overcome these critical obstacles.

#### 6.2.2. Immunotherapy: Vaccines and Antibodies Targeting ApoE4 and Amyloid-Beta

Another promising method for removing pathogenic aggregates in AD is immunotherapy. Several monoclonal antibodies targeting amyloid-beta (anti-Aβ), including aducanumab, lecanemab, have been shown to have sufficient ability during clinical trials to reduce amyloid-beta plaque levels and slow cognitive deterioration [37,49,92,93]. This immune response is only possible because these antibodies bind to the aggregated forms of Aβ and thus assist the microglia in its recognition and elimination. The antibodies developed for ApoE are another strand of immunotherapy research. These anti-ApoE antibodies make it possible to block the native interaction with amyloid-beta and thereby reduce the formation of plaques and consequently the associated neuroinflammation [85,87,94]. These results have been proven in preclinical studies, where this approach not only attenuated amyloid pathology but also improved cognitive function. Vaccines to induce an immune response against amyloid-beta protein and tau are being explored and possibly developed. However, even though the first vaccination attempts faced problems due to cross-reactivity and autoimmune reactions, the second generation of more specific and better-tolerated vaccines is currently being developed.

#### 6.2.3. Neuroprotective Strategies: Novel Agents Promoting Neuronal Survival and Synaptic Health

Neuroprotective agents aim to increase the capacity for resistance and neuronal adaptation in the face of injury or stress and to preserve synaptic function. Antioxidants like vitamins C and E may lower oxidative stress levels and prevent the formation of Aβ plaques. As such, these antioxidant agents protect neurons from damage while preserving neuronal plasticity [95,96,97]. Polyphenols such as resveratrol have antioxidant and anti-inflammatory properties, acting as potential neuroprotective agents. Ferroptosis is an iron-dependent method of programmed cell death. Strategies such as ferroptosis inhibitors act to inhibit the production of peroxidable lipids and the degradation of reactive oxygen compounds, helping to prevent cell death and slowing down neuronal degeneration [98]. Compounds that improve neurotrophic support, such as neurotrophic brain growth factor (BDNF) mimetics, can improve neuronal viability and synaptic plastics. In addition, lifestyle interventions such as physical activity, cognitive training, and dietary modifications can aid in neuroprotection. Regular physical exercise correlates with a boost in neurogenic activity and improved cognitive functions, while diets high in antioxidants or omega-3 fatty acids also benefit brain health [68].

**Table 3 cells-14-00243-t003:** Current therapies for Alzheimer’s disease (AD).

Therapeutic Strategy	Description of the Approach	Mechanism of Action	References
ApoE modulation	Small molecule and antisense oligonucleotide interventions	Alter the structure and function of ApoE4 by converting it to less toxic isoforms (ApoE3 or ApoE2)	[66,91]
Immunotherapy	Anti-Aβ and anti-ApoE monoclonal antibodies	Inhibit the formation of toxic aggregates of Aβ and disrupt the binding between ApoE and Aβ	[49,66]
Microglial activation	Nonsteroidal anti-inflammatory drugs (NSAIDs) and immune modulators	Reduction in neuronal inflammation, decreasing pro-inflammatory cytokines and favoring the microglial anti-inflammatory phenotype	[62,79]
Neuroprotective agents	Antioxidants (vitamin C, E), polyphenols (resveratrol), and ferroptosis inhibitors	Reduction in oxidative stress, protecting neurons from potential damage	[96]
Gene editing	CRISPR/Cas9 technology	Conversion of the E4 allele to E3 or E2 alleles, reversing degenerative effects associated with E4	[97]

This table demonstrates some of the current therapies used in Alzheimer’s disease (AD), providing a brief description of the approach and the associated mechanism of action.

### 6.3. Future Research Directions

#### 6.3.1. Biomarkers and Diagnostics: Identifying Early Biomarkers for Risk Assessment and Disease Monitoring

Early detection of AD is critical for effective intervention. Improvements in imaging methods, such as positron emission tomography (PET) with the use of amyloid and tau ligands, can identify pathological alterations leading clinical symptoms by years [62,99,100]. Magnetic resonance imaging (MRI) can quantify and assess the structural changes in the disease progression and brain atrophy.

Biochemical signatures, including the measurements of amyloid-beta, phosphorylated tau, and ApoE4 expression in the cerebrospinal fluid and blood provide valuable information for diagnosis [51,101]. Genetic screening for genotypes involved in the high risk of the pathology, such as ApoE4, can also identify individuals who could benefit from early interventions. The establishment of fully confident and non-invasive biomarkers is crucial for risk assessment, monitoring treatment efficacy, and advancing personalized medicine strategies.

#### 6.3.2. Personalized Medicine: Tailoring Treatments Based on Individual ApoE Genotype

Personalized medicine aims to personalize healthcare by employing individual genetic profiles, lifestyle habits, and environmental conditions. In the context of Alzheimer’s disease, tailoring treatments to an individual’s ApoE genotype can boost therapeutic effectiveness and lessen adverse reactions. Specific interventions based on the Apolipoprotein E (ApoE) genotype may involve structural and functional modification of ApoE4 using small molecules or gene editing technologies, already mentioned above as possible therapeutic strategies [11]. Personalized immunotherapies could be developed by targeting specific pathological characteristics present in patients, allowing for more individual and personalized treatment. The inclusion of diagnostic biomarkers to therapeutic strategies already in use allows interventions to be more precise, optimizing outcomes for patients with a specific pathology

#### 6.3.3. Broadening Scope: Investigating ApoE4’s Role in Other Neurodegenerative and Neuroinflammatory Diseases

Expanding research to explore the role of ApoE4 in other neurodegenerative and neuroinflammatory conditions could provide insights applicable to a wide range of neurological disorders. It has been established through various studies that ApoE4 drives the progression of Parkinson’s disease, multiple sclerosis, and amyotrophic lateral sclerosis [102]. An in-depth understanding of the common mechanisms by which this E4 allele influences neuroinflammation and neurodegeneration can guide the development of broad-spectrum therapeutic strategies. Additionally, the investigation of the complex interaction between ApoE4 and microglia in different disease contexts can identify new therapeutic targets to modulate the immune response and induce neuronal protection.

## 7. Conclusions

Apolipoprotein E4 is a key factor in the onset and progression of AD, with its influence extending beyond microglia to also include astrocytes and oligodendrocytes. These interactions disrupt essential processes such as lipid metabolism, myelination, amyloid-beta clearance, and neuronal communication, collectively worsening neurodegeneration. The novel insights presented in this review emphasize the importance of targeting ApoE4’s multifaceted effects to mitigate its contribution to AD pathology.

Emerging therapeutic strategies, including gene-editing technologies, antisense oligonucleotides, and immune modulation therapies, represent promising approaches to counteract the detrimental effects of ApoE4. Additionally, precision medicine tailored to individual ApoE genotypes presents new opportunities to personalize treatment and improve patient outcomes. This review also highlights the broader significance of ApoE4 in other neurodegenerative diseases, which could lead to cross-disciplinary therapeutic advantages.

Future research should focus on unraveling the complex interactions between ApoE4 and glial cells to identify actionable therapeutic targets. Furthermore, efforts to integrate early intervention strategies and public health initiatives—such as promoting healthy lifestyles and raising awareness of modifiable risk factors—will be crucial in preventing or delaying the onset of AD. Advancing our understanding of ApoE4’s roles across various glial cell types and neurodegenerative conditions has the potential to transform treatment paradigms and improve the quality of life for patients and their families.

## Figures and Tables

**Figure 1 cells-14-00243-f001:**
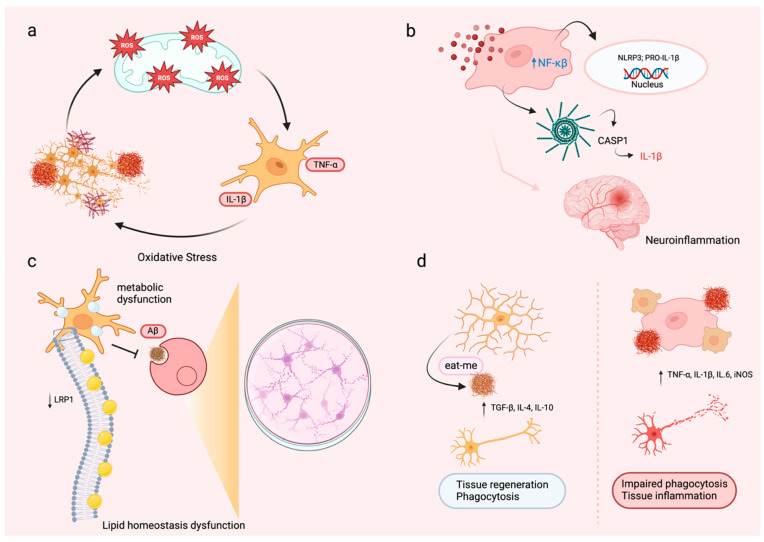
Multifaceted roles of ApoE4-induced Glial dysfunction in Alzheimer’s disease pathology. (**a**) Interaction between Aβ and oxidative stress in AD pathology: This panel emphasizes the detrimental effects of Aβ aggregation on oxidative stress in APoE4+ microglia. The presence of amyloid plaques generates excessive reactive oxygen species (ROS) in ApoE4+ microglia, which damage mitochondria and exacerbate oxidative stress in the brain. This promotes further Aβ aggregation and increases neurotoxicity, establishing a vicious cycle of neuronal damage and plaque accumulation. (**b**) Role of the NLRP3 inflammasome in neuroinflammation: This panel outlines the molecular pathways linking ApoE4 to inflammasome activation in microglia. Inflammatory stimuli activate the NF-κB signaling pathway, leading to the transcription of pro-inflammatory genes, including NLRP3 and pro-IL-1β. The NLRP3 inflammasome subsequently activates caspase-1 (CASP1), which converts pro-IL-1β into its active form, IL-1β. This perpetuates chronic neuroinflammation and contributes to neuronal damage. (**c**) Lipid metabolism dysfunction and impaired Aβ clearance: This panel illustrates how ApoE4-driven lipid metabolism dysfunction prevents astrocytes and microglia from maintaining homeostasis. Disrupted interactions with lipid transport receptors, such as LDL receptor-related protein 1 (LRP1), lead to intracellular lipid accumulation, compromising cellular integrity and reducing Aβ clearance. This dysfunction aggravates amyloid plaque buildup and neuronal degeneration. (**d**) Comparison of healthy and dysfunctional microglia in phagocytic capacity: Healthy microglia support tissue homeostasis by effectively phagocytizing Aβ and releasing anti-inflammatory cytokines like TGF-β, IL-4, and IL-10, aiding tissue repair. In contrast, ApoE4-expressing microglia demonstrate lower phagocytic activity, resulting in the accumulation of Aβ and cellular debris. Simultaneously, they secrete pro-inflammatory cytokines, including TNF-α, IL-1β, and IL-6, which promote chronic inflammation and neurodegeneration.

## Data Availability

No new data were created or analyzed in this study.

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
