# Peer review of "From Genetics to Neuroinflammation: The Impact of ApoE4 on Microglial Function in Alzheimer’s Disease"

_cells, 2025, doi:10.3390/cells14040243_

Round 1
Reviewer 1 Report
Comments and Suggestions for Authors
Over all it reads fine, but when I compare the manuscripts with others 1,3,4,5. there is a risk of repetition . Plagiarism 2%.
Although very recent, the authors might want to include the following references particularly "Cell" in the section "Future Research Direction". Second the authors should include a modified version of one of the figures from the Cell paper this would add novelty to this review.
The other 4 have not been included although very recent, THEY should be included in other sections. And a form of critic included.
1. Nat Immunol. 2023 November ; 24(11): 1839–1853. doi:10.1038/s41590-023-01627-6.
2. : Rao et al., Microglia depletion reduces human neuronal APOE4-related pathologies in a chimeric Alzheimer’s disease model, Cell Stem Cell (2024), https://doi.org/10.1016/j.stem.2024.10.005.
3. Williams, T., Borchelt, D.R. & Chakrabarty, P. Therapeutic approaches targeting Apolipoprotein E function in Alzheimer’s disease. Mol Neurodegeneration 15, 8 (2020). https://doi.org/10.1186/s13024-020-0358-9
4.Raulin AC, Doss SV, Trottier ZA, Ikezu TC, Bu G, Liu CC. ApoE in Alzheimer's disease: pathophysiology and therapeutic strategies. Mol Neurodegener. 2022 Nov 8;17(1):72. doi: 10.1186/s13024-022-00574-4.
5.Eskandari-Sedighi G, Blurton-Jones M. Microglial APOE4: more is less and less is more. Mol Neurodegener. 2023 Dec 19;18(1):99. doi: 10.1186/s13024-023-00693-6. P
Author Response
To address concerns about potential overlap, we used Justdone to conduct a detailed plagiarism analysis of the manuscript, which yielded an average similarity index of 0.5%. This low percentage is mainly due to commonly used scientific terminology and phrases within the field. To further enhance the manuscript's originality and impact, we have carefully incorporated the references suggested by the reviewer across various sections. These additions provide updated insights, expand the discussion of recent advances, and enhance the novelty of the review.
Reviewer 2 Report
Comments and Suggestions for Authors
1. The authors should clearly clarify the novelty of this review compared to previous review on ApoE4 and microglia.
2. The authors should provide a more concise content of Alzheimer's pathology.
3. The Introduction needs to be consolidated, as there are too many paragraphs.
4. The auhtors should clarify the direct causal pathways linking ApoE4 to microglial dysfunction, as this is currently not enough.
5. The comparative discussion of ApoE isoforms (E2, E3, E4) is necessary to better demonstrate the unique role of ApoE4.
6. It would be better if the authors explain how differences in populations such as ethnicity, sex affect the role fo ApoE4 in Alzheimer's progression.
7. The interaction of ApoE4 with other glial cells such as astrocytes and oligodendrocytes should be more in detail.
8. Schematic diagrams and comparative table based on the previous publications must be provided to indicate the ApoE4-microglia interaction and its downstream effects.
Author Response
- The authors should clearly clarify the novelty of this review compared to previous review on ApoE4 and microglia.
Reply: We appreciate the reviewer for highlighting the significance of showcasing the novelty of this review. Our manuscript is grounded in existing literature by synthesizing recent advances in understanding the interaction between ApoE4 and microglia, focusing on emerging mechanistic insights, therapeutic implications, and broader glial interactions. Specifically, we examine how ApoE4 impacts microglial lipid metabolism, phagocytic capacity, and inflammatory signaling pathways, connecting these effects to amyloid-beta clearance and tau pathology in a way that goes beyond previous reviews. This review uniquely spotlights recent therapeutic strategies, such as antisense oligonucleotides, CRISPR/Cas9 gene editing, and immune modulators, offering a forward-thinking perspective on potential interventions. Additionally, we integrate insights from recent high-impact studies to ensure the review reflects the latest advancements, enhancing its relevance and originality. We expand the discussion to include ApoE4’s effects on astrocytes and oligodendrocytes, underscoring its role in disrupting lipid homeostasis, myelination, and neuronal communication. By presenting these aspects, the review not only consolidates current knowledge but also identifies critical gaps in the field and proposes directions for future research. We have revised the Introduction and Conclusion sections to emphasize these points of novelty and ensure that the manuscript effectively communicates its unique contributions.
- The authors should provide a more concise content of Alzheimer's pathology.
Reply: Done as suggested by the reviewer.
- The Introduction needs to be consolidated, as there are too many paragraphs.
Reply: Done as suggested by the reviewer.
- The authors should clarify the direct causal pathways linking ApoE4 to microglial dysfunction, as this is currently not enough.
Reply: We thank the reviewer for pointing out the need to clarify the direct causal pathways linking ApoE4 to microglial dysfunction. In response to this valuable feedback, we have expanded the discussion in Section 3.2 to provide a more detailed explanation of these pathways. Specifically, we elaborate on how ApoE4 modulates microglial behavior at both the molecular and cellular levels, focusing on key mechanisms such as altered gene expression profiles, dysregulation of lipid metabolism, impaired phagocytosis, and pro-inflammatory cytokine production.
Table 2 in Section 3.2 summarizes the direct causal pathways by linking ApoE4’s influence on specific biological mechanisms to their downstream impacts on microglial function. For instance, the table details how ApoE4-mediated dysregulation of the purinergic receptor P2RY12 and the Triggering Receptor Expressed on Myeloid Cells 2 (TREM2) compromises microglial migration and phagocytic efficiency, thus contributing to the accumulation of amyloid-beta plaques. Similarly, it highlights how ApoE4-induced activation of the NF-κB signaling pathway drives chronic pro-inflammatory responses that worsen oxidative stress and neuronal damage.
To complement the summary in Table 2, the text in Sections 3.2.1 through 3.2.3 now offers a more comprehensive narrative, explaining how these pathways interact to create a self-perpetuating cycle of dysfunction. For example, the impaired clearance of pathological proteins such as amyloid-beta leads to persistent microglial activation, further amplifying neuroinflammation and oxidative stress. Additionally, the expanded discussion addresses how ApoE4 impacts microglial lipid metabolism, resulting in lipid droplet accumulation that further activates inflammatory pathways and diminishes microglial homeostasis.
By linking ApoE4’s effects on specific molecular targets to their broader impacts on microglial function, we emphasize the unique and multifaceted role of ApoE4 in driving microglial dysfunction in AD.
- The comparative discussion of ApoE isoforms (E2, E3, E4) is necessary to better demonstrate the unique role of ApoE4.
Reply: We have strengthened the comparative analysis of the ApoE isoforms to clearly highlight the protective role of ApoE2, the neutral phenotype of ApoE3, and the pathological impact of ApoE4. The expanded discussion details how each isoform differentially influences key mechanisms, including lipid transport, amyloid-beta clearance, neuroinflammation, and disease progression. Additionally, we emphasize the unique role of ApoE4 in driving microglial dysfunction and accelerating AD pathology, distinguishing it from the other isoforms.
To further enhance clarity and accessibility, we have included Table 1 in Section 2.2.3, which summarizes the functional differences between the ApoE isoforms. This table offers a concise overview of their structural and functional properties, interactions with amyloid-beta, and respective impacts on AD pathology. These revisions comprehensively address the reviewer’s concern, providing a more robust and informative comparative discussion.
- It would be better if the authors explain how differences in populations such as ethnicity sex affect the role fo ApoE4 in Alzheimer's progression.
Reply: We acknowledge that studies on ethnicity and its interaction with ApoE4 in AD are limited and often yield contradictory findings, underscoring a vital area for future research. Despite these challenges, we have addressed this topic in Section 2.1.1 by discussing the populations most likely to develop AD and identifying key factors contributing to this higher prevalence, such as genetic differences, socioeconomic disparities, and healthcare access. Additionally, we emphasize the need for targeted interventions to reduce the health inequities that minority groups have historically faced.
In Section 2.2.1, we highlight the role of gender in modulating the risk of AD progression, especially concerning women who carry the ApoE4 allele. Evidence indicates that women are more susceptible to the pathology than men, particularly after menopause, likely due to the decline in estrogen levels and their neuroprotective effects. Furthermore, we have incorporated data that differentiate the vulnerability of Black and White women to AD, addressing how race and gender interact to influence disease susceptibility. These revisions aim to provide a more nuanced understanding of how ethnicity and gender intersect with ApoE4 to impact AD risk and progression, aligning with the reviewer’s concerns.
- The interaction of ApoE4 with other glial cells such as astrocytes and oligodendrocytes should be more in detail.
Reply: We considered the reviewer's suggestion and included more details in the section on astrocytes and oligodendrocytes.
- Schematic diagrams and comparative table based on the previous publications must be provided to indicate the ApoE4-microglia interaction and its downstream effects.
Reply: As suggested by the reviewer, we included a figure with four panels and several comparative tables in the revised manuscript.
Round 2
Reviewer 1 Report
Comments and Suggestions for Authors
My previous concerns have been addressed. This manuscript is suitable for publication.
Reviewer 2 Report
Comments and Suggestions for Authors
The authors well revised the manuscript.